# Enhancing Milk Quality and Antioxidant Status in Lactating Dairy Goats through the Dietary Incorporation of Purple Napier Grass Silage

**DOI:** 10.3390/ani14050811

**Published:** 2024-03-06

**Authors:** Narawich Onjai-uea, Siwaporn Paengkoum, Nittaya Taethaisong, Sorasak Thongpea, Pramote Paengkoum

**Affiliations:** 1Department of Animal Science, Faculty of Agriculture at Kamphaeng Saen, Kasetsart University, Kamphaeng Saen Campus, Nakhon Pathom 73140, Thailand; 2Program in Agriculture, Faculty of Science and Technology, Nakhon Ratchasima Rajabhat University, Muang, Nakhon Ratchasima 30000, Thailand; siwaporn.p@nrru.ac.th; 3School of Animal Technology and Innovation, Institute of Agricultural Technology, Suranaree University of Technology, Muang, Nakhon Ratchasima 30000, Thailand; iszy.nittaya@gmail.com (N.T.); sorasak.t@sut.ac.th (S.T.)

**Keywords:** milk quality, antioxidant status, lactating dairy goats, dietary incorporation, purple Napier grass silage

## Abstract

**Simple Summary:**

The condition of oxidative stress is initiated by an imbalance between oxidants and antioxidants; subsequent interactions within the organism amplify this condition, leading to harmful effects on cell components. Anthocyanins, recognized for their antioxidant properties, play a significant role in producing various compounds, including polyphenols. The increased replacement of purple Napier grass silage has led to significant differences in lactose levels, the somatic cell count, glutathione S-transferase, the total antioxidant capacity, superoxide dismutase, 2,2-diphenyl-1-picrylhydrazyl radical, and the composition of anthocyanins. Consuming feed enriched with these bioactive compounds naturally and effectively maintains a healthy antioxidant balance. Additionally, this approach not only improves animal performance, but also ensures the safety of certain animal products which are intended for human consumption.

**Abstract:**

Oxidative stress resulting from an imbalance between oxidants and antioxidants can cause damage to certain cellular components. Purple Napier grass, a semi-dwarf variety, is characterized by its purple leaves and contains anthocyanins, which provide it with antioxidant properties. This study examined the effects of feeding purple Napier grass (“Prince”) silage to lactating dairy goats on blood antioxidant activity, milk yield, and milk quality. Eighteen female Saanen crossbred goats, weighing 52.34 ± 2.86 kg and producing milk for 14 ± 2 days, were systematically divided into three groups based on their lactation period in the previous cycle as follows: early, mid, and late lactation. In a randomized complete block design (RCBD), treatments were randomly allocated to six animals in each block. The dairy goats were fed a total mixed ration (TMR) consisting of the three following treatments: control (100% Napier Pakchong 1 grass silage), 50% (a 50% replacement of the control with purple Napier grass silage), and 100% (100% purple Napier grass silage). The results show that goats who were fed a diet including 100% purple Napier grass silage showed higher levels of certain milk contents, especially with regard to lactose, when compared to those who were fed a control diet, as well as a diet with a 50% replacement of purple Napier grass silage. The somatic cell count (SCC) of these goats was reduced. In terms of antioxidant activity, dairy goats who were fed 100% purple Napier grass silage showed higher levels of enzymes in both plasma and milk, including glutathione s-transferase, total antioxidant capacity, superoxide dismutase, and 2,2-diphenyl-1-picrylhydrazyl radical, compared to the control group and the 50% replacement group. The plasma and milk of these goats showed lower levels of malondialdehyde. The dairy goats who were fed a 100% purple Napier grass silage diet showed higher concentrations of anthocyanins, including C3G, P3G, Peo3G, M3G, Cya, Pel, and total anthocyanins in milk, when compared to the control group and the 50% replacement group. The increased replacement of purple Napier grass silage led to significant differences in lactose levels, somatic cell count, glutathione S-transferase, total antioxidant capacity, superoxide dismutase, 2,2-diphenyl-1-picrylhydrazyl radical, and the composition of anthocyanins. This study provides evidence to support the use of purple Napier grass silage as a beneficial source of roughage for lactating dairy goats.

## 1. Introduction

The electron transport chain of the mitochondria generates reactive oxygen species (ROS) during biochemical processes such as respiratory rupture caused by phagocytosis [1]. These ROS play a role in the oxidation of biomolecules and lipids, particularly those that are sensitive to specific molecules [2]. Aerobic organisms depend on dietary antioxidants as part of their effective protection system against ROS activity [3,4]. The effects of oxidative stress can be triggered by imbalances between oxidants and antioxidants, which are increased through interactions within the organism. Purba [4] and Celi [5] showed that oxidative stress has the potential to cause oxidative damage to cell molecules. Moreover, oxidative stress is intricately associated with nearly all diseases and physiological stages experienced by dairy cows and goats, including critical periods like the peripartum phase [5,6,7]. This highlights the significance of exploring natural interventions. Plant extracts or natural molecules possessing antioxidant properties, sourced from a diverse array of plant species such as polyphenols [8,9,10], offer promising avenues for mitigating oxidative stress and promoting animal health.

The purple Napier grass (*Pennisetum purpureum* “Prince”), a semi-dwarf cultivar with purple foliage, was jointly developed by the United States Department of Agriculture (Washington, DC, USA) and the University of Georgia College of Agricultural and Environmental Sciences (Griffin, GA, USA). Purple Napier grass, imported by the Dairy Farming Promotion Organization of Thailand (DPO) and temporarily housed at the study site in Saraburi Province, Thailand, awaits transplantation. Notably, this grass species contains anthocyanins; these are plant pigments known for their diverse functions, such as serving as food coloring additives and potentially contributing to reductions in disease risk [11,12]. Research into the functional and biological properties of anthocyanins from purple corn has involved both in vitro and in vivo experiments with ruminant animals. These studies have investigated the antioxidant activity of purple corn anthocyanin. Lactating dairy goats fed purple corn stover silage showed higher levels of important antioxidant enzymes such as superoxide dismutase (SOD), glutathione peroxidase (GPX), and catalase (CAT) [13].

According to Tian et al. [14], adding anthocyanins from purple corn pigment to the diet can potentially maintain stable levels of unsaturated fatty acids (UFA) in milk during storage. Furthermore, Purba et al. [15] found that using *Piper betle* L. powder (PL) as a feed additive has the potential to improve ruminal fermentation and biohydrogenation while also decreasing methane production.

Purba et al. [16] also reported that incorporating 17.6 g/kg sunflower oil into the diet, along with a practical dose of flavonoids and essential oils from *Piper betle* L. leaves, could be an effective strategy for increasing milk yield production and improving milk composition, including an increase in conjugated linoleic acid (CLA) content. Hosoda et al. [17] found that feeding anthocyanin-rich corn (*Zea mays* L., Choko C922) silage to lactating dairy cows reduced AST activity and increased SOD activity in plasma, indicating a potential lowering effect on AST with enhanced SOD activity. Anthocyanins enhance antioxidant enzymes, reduce oxidative stress, activate the Nrf2/ARE pathway, and mitigate the ROS-inhibition of enzymes, ultimately preserving cellular integrity by reducing oxidative damage [17]. The application of bioactive forages provides a natural and viable alternative for promoting a beneficial antioxidant balance, improving animal performance, and ensuring the safety of animal products for human consumption. This method for sustainable agriculture is novel and environmentally responsible.

Furthermore, there has been little research on feeding purple forage to ruminant animals, so the effects of anthocyanin feeding on these animals are unclear. The aim of this study was to investigate whether the incorporation of purple Napier grass silage into the diet of lactating dairy goats could enhance their antioxidant status and milk quality.

## 2. Materials and Methods

### 2.1. Experimental Design, Animals, and Diets

The experiment included eighteen female crossbred Saanen lactating goats, weighing an average of 52.34 ± 2.86 kg. These goats were divided into three groups based on their lactation stage (1^st^ to 3^rd^) and parity (1^st^ to 3^rd^), and all had healthy and symmetrical udders. Six animals were randomly assigned to one of the treatments within each block, using a randomized complete block design (RCBD). The trial lasted 56 days, consisting of a 14-day adjustment period, followed by a 42-day measurement period, during which samples were collected at the end of each experimental week. The dairy goats received a mixed total feed ration (TMR) in three treatments as follows: control, consisting of 100% Napier Pakchong 1 grass silage; 50%, where 50% of the control was replaced with purple Napier grass silage; and 100%, consisting of 100% purple Napier grass silage.

The grass silage was prepared in 2019 at the Suranaree University of Technology’s (SUT) experimental goat farm in Nakhon Ratchasima, Thailand (14°5337.9 N, 102°01′22 N. Then, Napier Pakchong 1 (*Pennisetum purpureum* cv. Pakchong 1) and purple Napier (*Pennisetum purpureum* “Prince”) were harvested and cut into 20–30 mm pieces using a forage cutter (EUROX, Bangkok, Thailand) after a 60-day regrowth period. For optimal preservation, the forages were carefully packed into 200 L plastic drums with clamp lids and stored until the feeding trial started. These containers were kept at a constant room temperature of between 27 and 30 °C.

### 2.2. Animal Management

The animal management protocols used in this study are outlined in Vorlaphim et al. [18]. The animals were housed in clean and dry pens, prioritizing their comfort and well-being, and ensuring convenient access to water. The feeding occurred twice a day, at 0700 and 1600 h, with diets referred to as total mixed rations (TMR). This feeding method allowed for ad libitum intake while taking into consideration a 10% refusal rate. The formulation of TMR diets followed the guidelines of the National Research Council [19], with a particular emphasis on meeting the nutritional requirements of dairy goats weighing an average of 50 kg.

### 2.3. Feeds and Milk Sampling

The daily weighing of the provided and refused feed before the morning feeding facilitated the calculation of dry matter intake (DMI) throughout the experimental period. The milk yield from the dairy goats was recorded once a day at 1900 h using a portable milking machine (Condor Company, made in Reggio Emilia, Italy). The initially produced milk was excluded from the initial milking procedure. The milking routine involved preparing the machinery and cleaning the udder and teats with a 1.0% sodium hypochlorite solution, followed by the actual milking process. A vacuum of 36 kPa was utilized, with a pulsator operating at 120 cycles per minute and a 50:50 milk/rest ratio. Eighteen milk samples were collected in the afternoon during the final two days of the experiment, covering the 3rd to 8th weeks, with two days per collection period.

### 2.4. Blood Sampling

Eighteen blood samples were collected in the morning during the final two days of the experiment, covering from the 3^rd^ to the 8^th^ weeks, with two days allocated for each collection period. Additionally, blood samples (approximately 3 mL each) were collected from the jugular vein at 0, 2, and 4 h after feeding using Vacuette^®^ tubes (Greiner Bio-One, Greiner Bio-One GmbH, Bad Haller Str., Kremsmünster, Austria) containing K3-EDTA. After centrifugation at 4000 r/min at 4 °C for 15 min, performed with the SorvallTM LegendTM XT/XF Centrifuge Series (Thermo Fisher Scientific Pte Ltd., Waltham, MA, USA), the resulting plasma was carefully transferred to a 1.5 mL tube and stored at −20 °C for subsequent analysis.

### 2.5. Chemical Analysis of Feeds and Milk

The evaluation of the nutritive value of grass silage involved a systematic series of procedures. Initially, freeze-drying was applied to the grass silage, followed by grinding, which was completed to ease the passage through a 1 mm^2^ mesh screen. The resulting finely ground sample underwent a total chemical composition analysis. The total nitrogen (N) was determined with the Kjeldahl method, and the crude protein (CP) was calculated by multiplying the N content by 6.25, following prior studies [20,21,22]. The quantification of the ether extract (EE) and the ash contents followed the guidelines established by the AOAC [20]. The estimation of the neutral detergent fiber (NDF) and acid detergent fiber (ADF) followed the methods described by Van Soest et al. [23]. Following the milking sessions, the thorough mixing and collection of milk samples occurred, and the pH value of the milk was promptly determined using a portable pH meter (Mettler Toledo, Schwerzenbach, Switzerland).

The milk samples were divided into two portions. One portion was preserved with bronopol tablets (D&F Control System Inc., San Ramon, CA, USA) at 4 °C until the analysis of fat, protein, lactose, total solids (TS), solids-not-fat (SNF), and somatic cell count (SCC) using a MilkoScan analyzer (MilkoScanTMFT2, FOSS, Hillerod, Denmark) took place. Before the analysis, this portion underwent a 10 min incubation at 40 °C in a water bath. The other portion was stored at −20 °C for the subsequent analysis of antioxidant and anthocyanin activity. Following the methodology outlined by Hamzaoui et al. [24], the calculation of fat-corrected milk (3.5% FCM) for milk production was performed using the formula 3.5% FCM = Kilograms of milk yield × [0.432 + 0.162 × (fat%)].

### 2.6. Antioxidant Activity

The scavenging activity of 2,2-diphenyl-1-picrylhydrazine (DPPH) was analyzed using spectrophotometric methods [25,26] with minor modifications. This study utilized the stable free radical DPPH (Sigma-Aldrich, St. Louis, MI, USA, Pcode: 101845869). In a 1.5-mL tube, 50 μL of each plasma and milk sample was mixed with 1 mL of DPPH reagent methanol solution (25 μmol/L). The mixture was thoroughly shaken and incubated in the dark at room temperature for 30 min, followed by centrifugation at 4000 r/min at 4 °C for 10 min. We then transferred the supernatant to a 96-well plate with 200 μL per well, and measured the absorbance at 517 nm using a microplate reader (Epoch, BioTek, Luzern, Switzerland). The DPPH-scavenging activity was calculated as a percentage using the following formula:Scavenging activity (%) = [Ac − (As × 100)]/Ac
where Ac represents the control absorbance, and As represents the sample absorbance.

The enzyme activity of the total antioxidant capability (TAC), superoxide dismutase (SOD), glutathione S-transferase (GST), and malondialdehyde (MDA) was measured using Sigma-Aldrich kits, namely, MAK187 for TAC, 19160 for the SOD determination, CS0410 for GST, and MAK085 for MDA. Subsequently, the enzyme activity of all samples was determined using a microplate reader (Thermo Scientific™, Waltham, MA, USA).

### 2.7. Anthocyanin Composition

The concentration of anthocyanins was determined using a modified HPLC method, with the standard stock solution, calibration standard, and sample preparation for quality control adapted from prior methods [27,28]. For the grass silage specimen (50 g), the extraction was performed using a 1% hydrochloric acid (HCl) solution dissolved in a 95% methanol solution (15:85, *v*/*v*). After incubating at 50 °C for 24 h [29], the resulting supernatant was collected. Subsequently, the supernatant was filtered through a 13 mm 0.45 μm Nylon Syringe Filter (Xiboshi, TNL1345PP, Tianjin Fuji Science & Technology Co., Ltd., Tianjin, China) to analyze the composition of anthocyanins using high-performance liquid chromatography (HPLC; 1260 Infinity II LC, Agilent Technologies, Santa Clara, CA, USA).

The analysis of the anthocyanin composition in milk followed a modified method [30,31]. After adjusting milk samples to a pH of 4 with 1% hydrochloric acid, researchers subjected them to acetone/water liquid–liquid extraction (70:30, *v*/*v*). Centrifugation at 10,000 r/min at 4 °C for 15 min was then performed. Following a 4 h incubation at room temperature, they collected the supernatant for anthocyanin composition analysis. Specimen analysis utilized high-performance liquid chromatography (HPLC) with a diode array detector (DAD). The extraction of anthocyanin content utilized a C_18_ Symmetry column with acetonitrile (CH_3_CN) as component A and a mixture of 10% acetic acid, 5% CH_3_CN, and 1% phosphoric acid in deionized water as component B. The overall analysis duration exceeded 30 min, with a 5 min interval between injections. Additional parameters included maintaining the sample temperature at 4 °C, using a 20 μL injection volume, setting a flow rate of 0.8 mL/min, maintaining a column temperature of 25 °C, and fixing the DAD wavelength at 520 nm.

### 2.8. Statistical Analysis

A one-way ANOVA was used to compare the means within each dietary treatment, and the differences in means were examined using the Tukey adjustment. The effect of the diet on various factors, including feed intake, milk yield, milk composition, antioxidant activity in plasma and milk, and the anthocyanin composition in the milk of lactating dairy goats, was examined using repeated measures modeling with the PROC MIXED procedure of Statistical Analysis System 9.4 [32], using a randomized complete block design (RCBD). The somatic cell count data were transferred to logarithm (with logarithm base 10) before statistical procedures. The statistical model considered the fixed effect of diet along with the random effects of the block (where the number of lactations is blocked) within the diet and farm. Orthogonal contrasts were used to evaluate the control and treatment effects of the diet using the CONTRAST statement. To determine significant differences (*p* < 0.05) among treatments, the Tukey–Kramer test was used, following the approach described by Steel and Torrie [33].

## 3. Results

### 3.1. Feed Intake, Milk Yield

The feed intake and milk yield of the experimental diets in this study are presented in Table 1, providing the ingredient and nutrient composition. Notably, the purple Napier grass silage (PNS) treatment showed an anthocyanin content that was 100% higher than in the other treatments. Table 2 provides data on the feed intake, milk yield, and milk efficiency. Dairy goats fed the 100% PNS treatment showed significantly higher dry matter intake (DMI) when compared to other treatments (*p* < 0.05). Furthermore, the intake of feed increased linearly with the level of purple Napier grass silage replacement (*p* < 0.05).

### 3.2. Milk Composition

The milk compositions of the dairy goats in the current study, including pH, fat, protein, lactose, total solids (TS), solids-not-fat (SNF), and somatic cell count (SCC), are detailed in Table 3. There were no significant differences in milk pH among treatments. The lactose content in the purple Napier grass silage treatment increased significantly (*p* < 0.05) when compared to the other treatments, demonstrating a linear increase (*p* < 0.05). Conversely, the SCC of dairy goats fed the purple Napier grass silage treatment was notably lower (*p* < 0.05) than in the other treatments, exhibiting both linear and quadratic decreases (*p* < 0.05). These observed variations in parameters are attributed to the increased rate of replacement with purple Napier grass silage.

### 3.3. Antioxidant Activity in Plasma and Milk

The antioxidant activity in plasma and milk is presented in Table 4, which includes the anthocyanin intake and parameters such as the DPPH-scavenging activity, total antioxidant capacity (TAC), superoxide dismutase (SOD), glutathione S-transferase enzyme (GST), and malondialdehyde (MDA) in both the plasma and milk of the dairy goats in the current study. The anthocyanin intake of dairy goats fed with purple Napier grass silage was significantly higher (*p* < 0.05) when compared to the other treatments, and this parameter showed a linear increase (*p* < 0.05).

Dairy goats fed 100% purple Napier grass silage had higher plasma antioxidant activity parameters (*p* < 0.05) than those in the other treatments, including SOD and GST (measured 2 and 4 h after feeding, along with the mean value). These parameters also showed a linear increase (*p* < 0.05). In contrast, plasma MDA levels (measured 2 and 4 h after feeding, along with the mean value) were lower (*p* < 0.05) than those in the other treatments, and this parameter showed a linear decrease (*p* < 0.05).

Dairy goats fed 100% purple Napier grass silage had higher values (*p* < 0.05) for milk antioxidant activity parameters, such as DPPH, TAC, SOD, and GST, than those fed the other treatments. These parameters also increased linearly (*p* < 0.05). Furthermore, MDA levels in raw milk decreased linearly (*p* < 0.05), and were lower (*p* < 0.05) than those in the other treatments. The higher amount of the purple Napier grass silage replacement was responsible for the significant differences in these parameters.

### 3.4. Anthocyanin Composition in Milk

The anthocyanin composition in raw milk, including cyanidin-3-glucoside (C3G), pelargonidin-3-glucoside (P3G), delphinidin (Del), peonidin-3-O-glucoside (Peo3G), malvidin-3-O-glucoside (M3G), cyanidin (Cya), pelargonidin (Pel), malvidin (Mal), and total anthocyanin, is presented in Table 5. Dairy goats fed the 100% purple Napier grass silage treatment showed higher levels (*p* < 0.05) of C3G, P3G, and Pel in their milk when compared to the other treatments, and these anthocyanin compositions showed a linear increase (*p* < 0.05). Similarly, Peo3G, M3G, Cya, and total anthocyanin were higher in the milk of dairy goats fed the 100% purple Napier grass silage treatment (*p* < 0.05) when compared to the other treatments, and these anthocyanin compositions increased linearly (*p* < 0.05). It should be noted that the analysis for the Del and Mal compositions in milk could not be conducted. The observed significant differences in these parameters can be attributed to the increased level of the purple Napier grass silage replacement.

## 4. Discussion

### 4.1. Feed Intake, Milk Yield, and Composition

The PNS 100% treatment demonstrated elevated levels of chemical composition and anthocyanin when compared to other treatments. This can be attributed to the response of purple plants to environmental stress through the anthocyanin metabolic process, thereby enhancing plant nutrients [34]. Tian et al. [35] found that anthocyanin-rich purple corn stover silage had higher levels of dry matter (DM) and crude protein (CP) than sticky corn stover silage. In the case of dairy goats fed with purple Napier grass silage, the milk composition was higher due to the increased dry matter intake (DMI) and nutrient intake. It is interesting that both the type of feed and DMI not only affect the nutrient composition of milk, but also influence the metabolism in the animal body, thereby affecting the energy and nutritional requirements for milk synthesis [36,37].

Harvatine and Allen [38] observed that with an increase in dry matter intake (DMI), there was a corresponding rise in both milk yield and milk protein yield responses. This increased DMI response affected responses to fat-corrected milk, milk fat percentage, and milk fat yield, among other parameters. Additionally, the increasing DMI response played a role in impacting marginal milk and milk protein yields, concurrently influencing marginal milk fat yields. In the context of this experiment, the elevated level of the purple Napier grass silage replacement is expected to influence rumen fermentation, particularly propionic acid (C_3_), which serves as a precursor to the synthesis of milk lactose [39].

The most prevalent anthocyanin is cyanidin-3-glucoside, which is abundantly present in purple Napier grass cultivars. The chemical structure of anthocyanin is linked to a variety of sugar moieties, including glucose, galactose, rhamnose, xylose, and arabinose [40,41]. These sugars can undergo acylation with ruminal aromatic acids, resulting in anthocyanin breakdown. According to Tian et al. [13], anthocyanin-rich purple corn (*Zea mays* L.) stover silage from the TPSS group resulted in an increased lactose content in milk when compared to sticky corn stover silage (CSSS).

The breakdown of anthocyanins implies that these sugar moieties act as precursors for ruminal C_3_ synthesis and lactose synthesis, which occur through absorption in the digestive tract [42,43]. Genetic factors also play a role in milk composition, with a heritability value of 50%. The nutrient content variation in milk is affected by 50% because of dietary conditions and management procedures [44]. The yield and composition of dairy goat milk vary significantly depending on factors such as breed, parity, lactation stage, age, geographical area, season, diet, health, and goat management [43,45,46].

The reduction in the somatic cell count (SCC) in milk, associated with the increased use of purple Napier grass silage, suggests a potential positive correlation between antibacterial activity and elevated anthocyanin intake. Anthocyanins, recognized for their capacity to hinder bacterial cell wall processes and induce cytoplasmic leakage, significantly disrupt bacterial growth [47]. The effects on bacterial cell membranes result from a combination of antibacterial agents and interactions with hydrogen-binding membrane proteins or hydrophobic interactions [16,22,48]. Anthocyanins, through their interaction with the cell membrane structure and the ions necessary for protein stability, can bind or donate electrons at the membrane interface, thereby exhibiting antibacterial properties [17,43,49,50].

The membrane potential, which serves as the primary energy source for almost all chemical reactions in living cells, is critical to microbial cell maintenance and development. The respiratory chains in bacterial membranes include a primary dehydrogenase, coenzyme Q, as well as the various cytochromes b, c, o, and a [51]. Hellingwerf and Konings [51] found retrochalcones, especially licochalcone and echinatin, isolated from the roots of *Glycyrrhiza inflata*, to be oxygen absorption inhibitors in sensitive Gram-positive bacterial cells such as *Micrococcus luteus*.

They also prevented NADH oxidation in cell membrane formulations. Licochalcones inhibited NADH-cytochrome c reductase but had no effect on cytochrome c oxidase. There was no inhibition of NADH-FMN oxidoreductase or NADH-CoQ reductase. Licochalcones are thought to inhibit respiratory functions via the microbial respiratory electron transport chain, especially between CoQ and cytochrome c [51]. Peptidoglycan is a vital component of the cell wall, and its inhibition constitutes a primary mechanism of action for traditional antimicrobial drugs and flavonoids. Wu et al. [52] examined the kinetics of D-alanine-D-alanine ligase, which catalyzes the production of the peptidoglycan terminal dipeptide UDPMur-NAc-pentapeptide. The study found that quercetin and apigenin act as inhibitors [53]. The antibacterial activity affects the function of the bacterial cell wall and binds to directly produced peptidoglycan precursors (D-Ala-D-Ala). This binding forms a cap that blocks cross-linking in the polypeptide chain, thereby damaging the strength of the bacterial cell wall [54].

### 4.2. Antioxidant Activity in Plasma and Milk

The 100% purple Napier grass silage treatment resulted in elevated levels of DPPH, TAC, SOD, and GST enzymes in both plasma and milk. Additionally, this treatment was associated with reduced levels of MDA, a correlation linked to the increased intake of anthocyanins. The two following main categories of antioxidants modulate free radical reactions: enzymatic antioxidants and non-enzymatic antioxidants. Anthocyanins, typically flavonoids, can donate electrons in their native forms to reactive oxygen species (ROS), effectively preventing the oxidation of biomolecules such as polyunsaturated fatty acids (PUFAs), proteins, and DNA. Furthermore, anthocyanins possess metal ion-chelating properties and act as free radical scavengers [4,16].

The radical-scavenging activity of anthocyanidins is intricately linked to specific structural features. Notably, the ortho-dihydroxy arrangement in the B-ring, the conjugated double bond at positions 2 and 3, and the presence of a 4-oxo function in the C-ring play pivotal roles in influencing this activity [55]. Additionally, flavonoids engage in the formation of metal ion complexes, utilizing hydroxyl and keto-substituents, or hydroxyl groups at positions 3 or 5 of the B-ring [55]. The analyses of structural interactions indicate that hydroxylation at positions C3′ and C5′ enhances hydrogen donation efficiency, highlighting the significant involvement of the B-ring in electron donation [56]. The oxygen radical-absorbing capacity (ORAC) among anthocyanin aglycones, the non-sugar components, exhibits variation. Specifically, the hydroxylation sequence in the A and C rings, along with enhanced hydroxylation in the B-ring, correlates with increased antioxidant potential. For instance, 3′, 4′ di-OH demonstrates a superior ORAC capacity when compared to 3′-OH [57].

Cyanidin exhibits a higher oxygen radical-absorbing capacity (ORAC) when compared to malvidin and peonidin. An interesting exception is observed with delphinidin. Despite having three hydroxyl groups on the C ring, it shows low antioxidant potential. This deviation can be attributed to the potential diminishing effect of the 5′-OH in the presence of 3′, 4′-OH (delphinidin) when compared to the presence of 3′, 4′-OH only (cyanidin) [57]. In parallel with anthocyanidins like cya-nidin-3-glucoside, hydroxyl groups bonded to phenolic rings have the ability to donate an electron, supplemented by a hydrogen nucleus, to free radicals [58,59].

Electrons stabilize and immobilize free radicals. During this process, the polyphenol-reducing agent transforms into the aroxyl radical, which is relatively more stable due to resonance when compared to the reduced free radical. As a result, harmful oxidative chain reactions are terminated [58]. It has been observed that anthocyanidins exhibit greater radical-scavenging capacity than anthocyanins, and this scavenging ability decreases with an increase in the number of sugar moieties [60].

The Nrf2/ARE pathway potentially serves as an additional mechanism by which anthocyanins augment the activities of SOD and GST enzymes within cells [61,62]. Nrf2, or nuclear factor E2-related factor 2, is the key regulator responsible for activating the antioxidant response element (ARE). Typically, Nrf2 is bound to Keap1 (Kelch-like erythroid CNC homologue (ECH)-associated protein 1). Reactive oxygen species (ROS) and reactive nitrogen species (RNS) are natural free radicals present throughout the body, arising as the by-products of various metabolic and immunological processes, including superoxide (O_2_^−^•), hydrogen peroxide (H_2_O_2_), hydroxyl (•OH), ozone (O_3_), and singlet oxygen [63].

The interaction between reactive oxygen species (ROS) and cysteine bonds on the Keap1 protein leads to the release of Nrf2 from Keap1. Subsequently, liberated Nrf2 translocates into the nucleus, initiating the activation of the antioxidant response element (ARE). This activation, in turn, enables ARE to prompt the synthesis of messenger RNA, which is responsible for encoding antioxidant enzymes, peptides, and proteins. Importantly, Nrf2 combines with antioxidants such as phenolic compounds, especially anthocyanins, in addition to responding to reactive oxygen species (ROS) and other pro-oxidant levels. The enzymes superoxide dismutase (SOD) and glutathione S-transferase (GST) play a key role in scavenging free radicals (ROS). Specifically, the enzyme SOD, located within both the cytosol and mitochondria, enhances the conversion of O_2_^−^• to oxygen and H_2_O_2_ in the presence of co-factors such as metal ions—copper (Cu), zinc (Zn), or manganese (Mn) [64].

Glutathione (GSH) serves as a crucial cellular antioxidant. Enzymes such as glutathione peroxidase (GSH-Px) and glutathione S-transferase (GST) are distributed throughout almost every human tissue, within both the cytoplasm and extracellular space. These enzymes play a vital role in converting hydrogen peroxide (H_2_O_2_) into water molecules. Glutathione enzymes are highly effective antioxidants against H_2_O_2_ and fatty acid hydroperoxides [65,66]. Additionally, the enzyme peroxyredoxin catalyzes the reduction of H_2_O_2_, organic hydroperoxides, and peroxynitrite (ONOO^−^).

Antioxidant enzymes play a crucial role in reducing lipid hydroperoxide and H_2_O_2_ levels, preventing lipid peroxidation, and maintaining cell membrane integrity and function. This enzymatic activity increases DPPH and TAC levels, while decreasing the MDA level. According to Tian et al. [13], lactating dairy goats fed anthocyaninrich purple corn stover silage showed elevated levels of superoxide dismutase (SOD) in both plasma and milk compared to those fed sticky corn stover silage.

Tian et al. [67] reported significantly higher levels (*p* < 0.05) of DPPH-scavenging activity and SOD enzymes in the plasma of goats fed PC2 (sticky corn stover silage with 1 g/d commercial purple corn pigment) and AR (anthocyanin-rich purple corn stover silage) when compared to those who were fed NC (negative control, rice straw) and PC1 (positive control, sticky corn stover silage). The inclusion of PC2 and AR resulted in a significant increase (*p* < 0.05) in the abundance of nuclear factor (erythroid-derived 2)-like 2 (NFE2L2) and a decrease (*p* < 0.05) in the level of tumor necrosis factor in the mammary gland.

Goats that consumed anthocyanin-rich (AR) feed tended to show elevated levels of SOD2, GPX1, and GPX2 mRNA expression in their mammary gland (*p* < 0.05). Additionally, significant positive correlations (*p* < 0.05) were identified between DPPH-scavenging activity, total antioxidant capacity, plasma SOD, catalase enzymes, and the abundance of NFE2L2 in the mammary gland. These studies suggest that the anthocyanin present in purple Napier grass silage could enhance the activity of SOD and GST enzymes, and it could also increase DPPH and TAC levels in dairy goats. Consequently, this enhancement of the antioxidant system may lead to a decreased concentration of MDA, thus preventing oxidative stress from affecting MDA by-products derived from DNA and lipid oxidation. It is important to note that the MDA concentration has primarily served as a measure of oxidant status [68].

### 4.3. Anthocyanin Composition in Milk

Dairy goats fed 100% purple Napier grass silage produced milk with higher anthocyanin concentrations, particularly cyanidin-3-glucoside (C3G), significantly surpassing other anthocyanins due to its positive correlation with anthocyanin intake. Anthocyanins, flavonoids, and secondary plant metabolites are abundant in fruits with colors ranging from red to blue, reaching concentrations of up to 1 g per 100 g of fresh weight [69].

The glycosylation property of anthocyanins likely affects the absorption mechanism, influencing the partitioning coefficients. These compounds passively diffuse through biological membranes, potentially distributing within various cellular phases. Aglycones, predominantly hydrophobic, easily diffuse through biological membranes via passive transport. However, when linked with sugars, their water solubility increases, reducing passive diffusion. The transportation of flavonoid glycosides occurs through two pathways. The first pathway features a sodium–glucose co-transporter (SGLT1 and GLUT2) which transports retained glucosides. The second pathway involves lactate phlorizin hydrolase at the brush border, facilitating extracellular glycoside hydrolysis, followed by the passive diffusion of the aglycone [70,71,72,73].

In the initial step, lactate phlorizin hydrolase is likely to hydrolyze anthocyanin glycosides in the mucosal brush border membrane [71,72]. The second absorption mechanism involves transferring retained anthocyanin glycosides to the enterocyte, which is possibly facilitated by a sodium–glucose co-transporter, as demonstrated for other flavonoids [72,74,75,76,77]. Once inside the cell, the glycoside can either directly cross the basolateral membrane and enter the portal circulation, or cytosolic β-glucosidase can hydrolyze it before intestinal metabolism and subsequent transport [72,74,76,78,79].

Lactate phlorizin hydrolase and cytosolic β-glucosidase do not metabolize cyanidin and delphinidin glucosides [80]. Despite this, cyanidin glycosides are absorbed and found in the bloodstream as intact glycosides and glucuronide derivatives, indicating that intact compound transport and hydrolysis precede transport [81,82,83,84,85]. In a 12-day study, Felgines et al. [69] investigated anthocyanin derivative ratios in rat organs after an anthocyanin-enriched blackberry diet. The bladder showed the highest anthocyanin concentrations, along with the prostate. Additionally, the prostate, testes, and heart store endogenous cyanidin-3-glucoside and a small amount of cyanidin monoglucuronide. The adipose tissue also contained cyanidin-3-glucoside and its methylated derivatives [69]. Tian et al. [13] found that anthocyanin-rich purple corn stover silage, containing abundant anthocyanins, transfers these compounds to milk, enhancing the antioxidant levels in milk from lactating dairy goats. This study provides evidence that anthocyanins undergo metabolic processes, as well as absorption, and manifest in the mammary gland, contributing to their presence in the milk of lactating dairy goats.

## 5. Conclusions

The results of this study showed that feeding dairy goats with 100% purple Napier grass silage improved various parameters. These enhancements consisted of improving milk composition (lactose), elevating the activity of antioxidants, and enhancing the anthocyanin composition in raw milk. In terms of plasma antioxidant activity, the 100% purple Napier grass silage treatment increased DPPH-scavenging activity, total antioxidant capacity (TAC), and plasma enzyme activity (SOD and GST) both 2 and 4 h after feeding. The dairy goats fed 100% purple Napier grass silage showed higher DPPH-scavenging activity, total antioxidant capacity (TAC), and antioxidant enzyme levels (SOD and GST), along with increased anthocyanin content in raw milk. In addition, dairy goats fed 100% purple Napier grass silage showed the lowest levels of malondialdehyde (MDA) in both plasma and milk. Based on the results of this study, it can be concluded that purple Napier grass silage serves as a beneficial roughage source for lactating dairy goats.

## Figures and Tables

**Table 1 animals-14-00811-t001:** Ingredient and nutrient composition of purple Napier grass silage substitution level in diets for dairy goats.

Item ^1^	Treatment ^2^		
	Control	50%	100%
Ingredient (%DM)	On dry basis%		
Napier Pak Chong 1 grass silage	50.00	25.00	-
Purple Napier grass silage	-	25.00	50.00
Soybean hull	4.50	4.50	4.50
Soybean residue	30.00	30.00	30.00
Concentrate 21% CP	15.00	15.00	15.00
Premix	0.50	0.50	0.50
Total	100.00	100.00	100.00
Chemical composition			
DM	36.27	35.97	35.67
	On dry basis%		
OM	90.86	89.96	89.24
CP	17.28	17.39	17.49
NDF	72.16	69.79	67.41
ADF	41.30	40.99	40.68
Hemicellulose	30.86	28.80	26.73
CF	22.71	22.07	21.42
EE	4.55	4.68	4.81
ME, kJ/g DM	2464.30	2497.76	2531.22
DE, kJ/g DM	3005.25	3046.05	3086.85
GE, kJ/g DM	3998.61	4082.64	4166.67
Anthocyanin (mg/kg DM)	446.92	813.69	1180.61

^1^ DM, dry matter; OM, organic matter; CP, crude protein; NDF, neutral detergent fiber; ADF, acid detergent fiber; CF, crude fiber; EE, ether extract; ME, metabolizable energy (ME = 0.82 × DE); DE, digestible energy; GE, gross energy. ^2^ Control, 100% Napier Pakchong 1 grass silage; 50%, control replaced with 50% purple Napier grass silage; 100%, 100% purple Napier grass silage.

**Table 2 animals-14-00811-t002:** Effects of the purple Napier grass silage replacement on DMI and milk yield in dairy goats.

Item ^1^	Treatment ^2^			SEM ^3^	*p*-Value ^4^		
	Control	50%	100%		T	Linear	Quadratic
Dry matter intake, g/d	1164.49 ^b^	1172.21 ^ab^	1214.87 ^a^	8.78	0.030	0.014	0.28
Milk yield, kg/d	1.20	1.25	1.40	0.041	0.12	0.050	0.55
3.5% FCM, kg/d	1.25	1.30	1.51	0.057	0.13	0.059	0.50
Fat, g/d	45.02	46.95	55.73	2.40	0.15	0.071	0.48
Protein, g/d	40.74	44.39	51.36	2.29	0.16	0.063	0.72
Lactose, g/d	53.50 ^c^	59.10 ^b^	74.85 ^a^	3.90	0.060	0.023	0.50
Total solid, g/d	139.26	150.44	181.93	8.56	0.10	0.042	0.55
SNF, g/d	94.24	103.49	126.20	6.18	0.087	0.034	0.58

^a,b,c^ Within a row, means without a common superscript differ significantly (*p* < 0.05). ^1^ 3.5% FCM, the fat-corrected milk; SNF, solids-not-fat. ^2^ Control, 100% Napier Pakchong 1 grass silage; 50%, control replaced with 50% purple Napier grass silage; 100%, 100% purple Napier grass silage. ^3^ SEM, standard error of mean. ^4^ *p*-value, control vs. treatment.

**Table 3 animals-14-00811-t003:** Effects of the purple Napier grass silage replacement on milk composition in dairy goats.

Item ^1^	Treatment ^2^			SEM ^3^	*p*-Value ^4^		
	Control	50%	100%		T	Linear	Quadratic
pH	6.57	6.58	6.60	0.012	0.65	0.36	0.92
Fat, %	3.73	3.74	3.95	0.062	0.29	0.16	0.48
Protein, %	3.38	3.53	3.64	0.057	0.18	0.069	0.82
Lactose, %	4.40 ^b^	4.68 ^b^	5.29 ^a^	0.12	0.001	0.0002	0.32
TS, %	11.50	11.95	12.87	0.27	0.098	0.037	0.66
SNF, %	7.78	8.21	8.93	0.21	0.068	0.023	0.73
SCC, cells × 10^6^/mL	1.93 ^a^	1.26 ^b^	1.14 ^c^	0.095	<0.0001	<0.0001	<0.0001

^a,b,c^ Within a row, means without a common superscript differ significantly (*p* < 0.05). ^1^ TS, total solid; SNF, solids-not-fat; SCC, somatic cell count. ^2^ Control, 100% Napier Pakchong 1 grass silage; 50%, control replaced with 50% purple Napier grass silage; 100%, 100% purple Napier grass silage. ^3^ SEM, standard error of mean. ^4^ *p*-value, control vs. treatment.

**Table 4 animals-14-00811-t004:** Effects of the purple Napier grass silage replacement on antioxidant activity in the plasma and milk of dairy goats.

Item ^1^	Treatment ^2^			SEM ^3^	*p*-Value ^4^		
	Control	50%	100%		T	Linear	Quadratic
Anthocyanin intake, mg/day	520.43 ^c^	953.81 ^b^	1434.29 ^a^	90.72	<0.0001	<0.0001	0.10
Plasma							
DPPH, %	24.00	25.03	26.64	0.70	0.32	0.14	0.85
TAC, nmole/µL							
0 h	1.04	1.06	1.07	0.025	0.91	0.68	0.92
2 h	1.12	1.15	1.19	0.026	0.50	0.25	0.93
4 h	1.08	1.11	1.14	0.024	0.63	0.35	0.94
Mean	1.08	1.11	1.14	0.025	0.67	0.38	0.99
SOD, %							
0 h	67.91	76.32	80.17	2.23	0.059	0.022	0.58
2 h	76.28 ^b^	86.63 ^a^	91.67 ^a^	2.22	0.004	0.001	0.43
4 h	75.01 ^b^	78.01 ^b^	82.81 ^a^	1.02	0.001	0.0002	0.48
Mean	73.06 ^b^	80.32 ^ab^	84.89 ^a^	1.75	0.008	0.003	0.63
GST, mmol/min/mL							
0 h	31.84	32.82	30.54	0.96	0.65	0.60	0.46
2 h	38.37 ^c^	51.99 ^b^	61.17 ^a^	2.74	<0.0001	<0.0001	0.40
4 h	34.73 ^c^	47.75 ^b^	56.20 ^a^	2.62	<0.0001	<0.0001	0.40
Mean	35.47 ^c^	43.82 ^b^	48.00 ^a^	1.48	<0.0001	<0.0001	0.10
MDA, umol/L							
0 h	0.79	0.80	0.82	0.021	0.88	0.62	0.94
2 h	0.71 ^a^	0.60 ^b^	0.50 ^c^	0.024	<0.0001	<0.0001	0.81
4 h	0.75 ^a^	0.68 ^b^	0.61 ^c^	0.018	<0.0001	<0.0001	0.78
Mean	0.75 ^a^	0.69 ^ab^	0.64 ^b^	0.017	0.012	0.004	0.88
Milk							
DPPH, %	25.33	29.91	32.00	1.23	0.066	0.025	0.59
TAC, nmole/µL	0.90	0.97	1.04	0.027	0.095	0.033	0.97
SOD, %	67.71 ^b^	68.70 ^b^	74.28 ^a^	0.96	0.002	0.001	0.10
GST, mmol/min/mL	33.40 ^c^	46.55 ^b^	55.48 ^a^	2.48	<0.0001	<0.0001	0.10
MDA, umol/L	0.50 ^a^	0.42 ^b^	0.37 ^c^	0.015	<0.0001	<0.0001	0.26

^a,b,c^ Within a row, means without a common superscript differ significantly (*p* < 0.05). ^1^ DPPH, 2, 2-diphenyl-1-picrylhydrazyl; TAC, total antioxidant capacity at 0, 2, and 4 h after feeding; SOD, superoxide dismutase (inhibition rate) at 0, 2, and 4 h after feeding; GST, glutathione S-transferase 0, 2, and 4 h after feeding; MDA, malondialdehyde 0, 2, and 4 h after feeding. ^2^ Control, 100% Napier Pakchong 1 grass silage; 50%, control replaced with 50% purple Napier grass silage; 100%, 100% purple Napier grass silage. ^3^ SEM, standard error of mean. ^4^ *p*-value, control vs. treatment.

**Table 5 animals-14-00811-t005:** Comparison of purple Napier grass silage level on the anthocyanin composition in milk.

Item ^1^	Treatment ^2^			SEM ^3^	*p*-Value ^4^		
	Control	50%	100%		T	Linear	Quadratic
C3G, mg/kg	1.00 ^c^	1.84 ^b^	2.61 ^a^	0.18	<0.0001	<0.0001	0.66
Del, mg/kg	-	-	-	-		-	-
P3G, mg/kg	0.54 ^c^	1.03 ^b^	1.55 ^a^	0.11	<0.0001	<0.0001	0.76
Peo3G, mg/kg	0.60 ^c^	1.05 ^b^	1.52 ^a^	0.10	<0.0001	<0.0001	0.82
M3G, mg/kg	0.23 ^c^	0.42 ^b^	0.61 ^a^	0.042	<0.0001	<0.0001	0.99
Cya, mg/kg	0.56 ^c^	1.02 ^b^	1.47 ^a^	0.10	<0.0001	<0.0001	1.00
Pel, mg/kg	0.60 ^c^	0.89 ^b^	1.30 ^a^	0.083	<0.0001	<0.0001	0.47
Mal, mg/kg	-	-	-	-		-	-
Total, mg/kg	3.53 ^c^	6.24 ^b^	9.06 ^a^	0.62	<0.0001	<0.0001	0.87

^a,b,c^ Within a row, means without a common superscript differ significantly (*p* < 0.05). ^1^ C3G, cyanidin-3-glucoside; Del, delphinidin; P3G, pelargonidin-3-glucoside; Peo3G, peonidin-3-O-glucoside; M3G, malvidin-3-O-glucoside; Cya, cyanidin; Pel, pelargonidin; Mal, malvidin (-, not detected). ^2^ Control, 100% Napier Pakchong 1 grass silage; 50%, control replaced with 50% purple Napier grass silage; 100%, 100% purple Napier grass silage. ^3^ SEM, standard error of mean. ^4^ *p*-value, control vs. treatment.

## Data Availability

The data presented in this study are available on request from the corresponding author. The data are not publicly available due to privacy or ethical restrictions.

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
