# Peer review of "Enhancing Milk Quality and Antioxidant Status in Lactating Dairy Goats through the Dietary Incorporation of Purple Napier Grass Silage"

_animals, 2024, doi:10.3390/ani14050811_

Round 1

Reviewer 1 Report

Comments and Suggestions for Authors

Manuscript is poorly written and need major modification before going to deep review. Before reviewing properly, I would like to invite authors to revise the manuscript as suggestion given below. Be careful in the abbreviations that authors made on several occasion and repeated in manuscript many times.

Line 16. ‘interactions within the organism’ what does interaction means?

Line 16: what do you mean by ‘this condition’

Line 17-18: rewrite the sentence ‘Anthocyanins, known for their antioxidant properties, contribute significantly to the production of various compounds, including polyphenols’

Line 19: in the previous sentence you are only mentioning polyphenols but here ‘these bioactive’ creating confusion. Please be specific

Line 19-20:  ‘healthy antioxidant balance’ in animals or what ?

Line 22-23: rewrite the sentence ‘The oxidative stress caused by an imbalance between oxidants and antioxidants damages cell components’

Line 23-24: sentence not clear ‘Anthocyanins and purple leaves characterize purple Napier grass, a semi-dwarf variety with antioxidant properties’

Line 25: do you mean milk quality ? if yes please mention ‘quality’

Line 26: do you mean milk in days? ‘milk for 14±2 days’ if yes, what were the production ?

Line 30: does ‘TMR’ represent total mix ration? If yes, abbreviate the term on first seen

Line 30 and 31: ‘Napier Pakchong 1 grass’ and ‘purple Napier grass silage’ are same or different?

Line 34-35: Rewrite ‘These goats also had a de- 34 creased somatic cell count (SCC)’

Line 36-37: what does ‘GST, TAC, SOD, and DPPH’ represents?

Line 37-38: ‘The plasma and milk of these goats showed lower levels of malondialdehyde (MDA)’ what does these goats represents?

Line 38: remove ‘(MDA)’

Line 42: ‘in these parameters’ please be specific

Line 55: ‘this stress’ be specific

Line 56: why ‘d’ is capital for ‘Diet’

Line 56: remove ‘all’

Line 56-57: remove the sentence ‘Diet, season, calving conditions, heat stress, and milk yield all have an effect on the delicate balance between pro-oxidants and antioxidant capacity’

Line 57-61: how can you correlate these two sentence in one paragraph ‘Furthermore, oxidative stress has been linked to almost all diseases and physiological stages in dairy cows and  goats, such as the peripartum period [5−7]. The extracts of plants or natural plant molecules with antioxidant properties are produced through a wide range of plant species, such as polyphenols [8−10]’

Line 63-65: please write full address of departments with country and country code  ‘….United States Department of Agriculture (USDA) and the University of Georgia College of Agricultural and Environmental Sciences (UGA)’

Line 65-68: how can you correlate these two sentences ‘Purple Napier grass, imported by the Dairy Farming Promotion Organization of Thailand (DPO), was stored at the study site in Saraburi Province, Thailand, until transplantation. Anthocyanins, pigments found in plants, serve a variety of functions, including food coloring additives and the potential reduction of disease risk [11,12]’

Line 69-70: incorrect sentence ‘Previous studies on the functional and biological properties of anthocyanins from purple corn have involved both in vitro and in vivo experiments with ruminant animals’ could you explain what kind of studies? I cant find any study here ?

Line 75-81: what is the relevance ‘According to Tian et al. [14], adding anthocyanins from purple corn pigment to the diet can potentially maintain stable levels of unsaturated fatty acids (UFA) in milk during storage. Furthermore, Purba et al. [15] found that using Piper betle L. powder (PL) as a feed additive has the potential to improve ruminal fermentation and biohydrogenation while decreasing methane production. Additionally, Purba et al. [16] showed that piper oil (PO), particularly at a dosage of 45 mg, is as effective as substitutes like sunflower oil  (SFO) in mitigating methane emissions without ad-versely affecting rumen fermentation”

Line 122: ‘mixed rations (TMR)’ has been abbreviated 107

Line 129: you are abbreviating dry matter intake (DMI) in subsequent sections, be carefull

Line 226-229: is this statistical data? ‘Notably, the purple Napier grass silage (PNS) treatment showed a chemical composition—crude protein (CP), ether extract (EE), metabolizable energy (ME), digestible energy (DE), gross energy (GE), and anthocyanin content—that was 100% higher than in the other treatments’

Comments on the Quality of English Language

poor 

Author Response

Response to Reviewer 1 Comments

Point 1: Line 16. ‘interactions within the organism’ what does interaction means?

Response 1: In the context of the sentence you provided, "interactions within the organism" refers to the various biochemical and physiological processes occurring within an organism's cells and tissues. These interactions involve the activities of different molecules, such as oxidants (reactive oxygen species) and antioxidants, which play crucial roles in maintaining cellular homeostasis. The imbalance between oxidants and antioxidants disrupts these interactions, leading to oxidative stress. This disruption can cause harmful effects on cell components, such as proteins, lipids, and DNA, potentially contributing to various diseases and conditions. Therefore, "interactions within the organism" refers to the dynamic biochemical processes that occur within cells and tissues, including the regulation of oxidative stress.

Point 2: Line 16: what do you mean by ‘this condition’

Response 2: In the context provided, "this condition" refers specifically to oxidative stress. Oxidative stress occurs when there is an imbalance between the production of reactive oxygen species (oxidants) and the body's ability to neutralize them with antioxidants. So, when the text mentions "interactions within the organism then lead to this condition," it's stating that the imbalance between oxidants and antioxidants within the organism's cells and tissues leads to oxidative stress. This oxidative stress, in turn, can cause harmful effects on various cell components, such as proteins, lipids, and DNA, as mentioned in the sentence.

Point 3: Line 17-18: rewrite the sentence ‘Anthocyanins, known for their antioxidant properties, contribute significantly to the production of various compounds, including polyphenols’

Response 3: Please see the attachment.

Point 4: Line 19: in the previous sentence you are only mentioning polyphenols but here ‘these bioactive’ creating confusion. Please be specific

Response 4: Please see the attachment.

Point 5: Line 19-20: ‘healthy antioxidant balance’ in animals or what ?

Response 5: The phrase "healthy antioxidant balance" refers to the state of having an appropriate level of antioxidants in the body to counteract oxidative stress and prevent cellular damage. In this context, it specifically pertains to animals that consume feed enriched with anthocyanins.

Point 6: Line 22-23: rewrite the sentence ‘The oxidative stress caused by an imbalance between oxidants and antioxidants damages cell components’

Response 6: Please see the attachment.

Point 7: Line 23-24: sentence not clear ‘Anthocyanins and purple leaves characterize purple Napier grass, a semi-dwarf variety with antioxidant properties’

Response 7: Please see the attachment.

Point 8: Line 25: do you mean milk quality ? if yes please mention ‘quality’

Response 8: Please see the attachment.

Point 9: Line 26: do you mean milk in days? ‘milk for 14±2 days’ if yes, what were the production ?

Response 9: Yes, "milk for 14±2 days" suggests that the goats were producing milk continuously for a period of approximately 14 days, with a possible variation of ±2 days. As for the production during this time period, it would depend on the specific data provided in the study.

Point 10: Line 30: does ‘TMR’ represent total mix ration? If yes, abbreviate the term on first seen

Response 10: Please see the attachment.

Point 11: Line 30 and 31: ‘Napier Pakchong 1 grass’ and ‘purple Napier grass silage’ are same or different?

Response 11: In the context provided, "Napier Pakchong 1 grass silage" and "purple Napier grass silage" are different types of silage made from different varieties of Napier grass.

- "Napier Pakchong 1 grass silage" refers to silage made from a specific variety of Napier grass called "Pakchong 1."

- "Purple Napier grass silage" refers to silage made from a variety of Napier grass that produces purple-colored leaves.

So, they are different types of Napier grass, each used to produce silage for different treatments in the study.

Point 12: Line 34-35: Rewrite ‘These goats also had a decreased somatic cell count (SCC)’

Response 12: Please see the attachment.

Point 13: Line 36-37: what does ‘GST, TAC, SOD, and DPPH’ represents?

Response 13: Please see the attachment.

Point 14: Line 37-38: ‘The plasma and milk of these goats showed lower levels of malondialdehyde (MDA)’ what does these goats represents?

Response 14: The phrase "these goats" refers to the dairy goats that were the subjects of the study or discussion. Specifically, it refers to the goats being described in the context, which showed lower levels of malondialdehyde (MDA) in both their plasma and milk.

Point 15: Line 38: remove ‘(MDA)’

Response 15: Please see the attachment.

Point 16: Line 42: ‘in these parameters’ please be specific

Response 16: Please see the attachment.

Point 17: Line 55: ‘this stress’ be specific

Response 17: Please see the attachment.

Point 18: Line 56: why ‘d’ is capital for ‘Diet’

Response 18: Please see the attachment.

Point 19: Line 56: remove ‘all’

Response 19: Please see the attachment.

Point 20: Line 56-57: remove the sentence ‘Diet, season, calving conditions, heat stress, and milk yield all have an effect on the delicate balance between pro-oxidants and antioxidant capacity’

Response 20: Please see the attachment.

Point 21: Line 57-61: how can you correlate these two sentence in one paragraph ‘Furthermore, oxidative stress has been linked to almost all diseases and physiological stages in dairy cows and  goats, such as the peripartum period [5−7]. The extracts of plants or natural plant molecules with antioxidant properties are produced through a wide range of plant species, such as polyphenols [8−10]’

Response 21: Please see the attachment.

Point 22: Line 63-65: please write full address of departments with country and country code  ‘….United States Department of Agriculture (USDA) and the University of Georgia College of Agricultural and Environmental Sciences (UGA)’

Response 22: Please see the attachment.

Point 23: Line 65-68: how can you correlate these two sentences ‘Purple Napier grass, imported by the Dairy Farming Promotion Organization of Thailand (DPO), was stored at the study site in Saraburi Province, Thailand, until transplantation. Anthocyanins, pigments found in plants, serve a variety of functions, including food coloring additives and the potential reduction of disease risk [11,12]’

Response 23: Please see the attachment.

Point 24: Line 69-70: incorrect sentence ‘Previous studies on the functional and biological properties of anthocyanins from purple corn have involved both in vitro and in vivo experiments with ruminant animals’ could you explain what kind of studies? I cant find any study here ?

Response 24: Please see the attachment.

Point 25: Line 75-81: what is the relevance ‘According to Tian et al. [14], adding anthocyanins from purple corn pigment to the diet can potentially maintain stable levels of unsaturated fatty acids (UFA) in milk during storage. Furthermore, Purba et al. [15] found that using Piper betle L. powder (PL) as a feed additive has the potential to improve ruminal fermentation and biohydrogenation while decreasing methane production. Additionally, Purba et al. [16] showed that piper oil (PO), particularly at a dosage of 45 mg, is as effective as substitutes like sunflower oil  (SFO) in mitigating methane emissions without adversely affecting rumen fermentation”

Response 25: The relevance of the provided information resides in its implications for animal nutrition and methane emissions reduction in dairy farming. Tian et al. [14] suggest that adding anthocyanins from purple corn pigment to the diet might maintain stable levels of unsaturated fatty acids (UFA) in milk during storage, potentially enhancing milk quality. Purba et al. [15] and [16] investigate the use of Piper betle L. powder (PL) and piper oil (PO) as feed additives, respectively, highlighting their potential to enhance ruminal fermentation, biohydrogenation, and reduce methane production. These studies highlight the importance of dietary strategies and supplements in enhancing both animal health and environmental sustainability within the dairy industry.

Point 26: Line 122: ‘mixed rations (TMR)’ has been abbreviated 107

Response 26: Please see the attachment.

Point 27: Line 129: you are abbreviating dry matter intake (DMI) in subsequent sections, be carefull

Response 27: Please see the attachment.

Point 28: Line 226-229: is this statistical data? ‘Notably, the purple Napier grass silage (PNS) treatment showed a chemical composition—crude protein (CP), ether extract (EE), metabolizable energy (ME), digestible energy (DE), gross energy (GE), and anthocyanin content—that was 100% higher than in the other treatments’

Response 28: No, the statement provided does not contain statistical data. Instead, it presents a comparison of the chemical composition of the purple Napier grass silage (PNS) treatment with other treatments. The comparison indicates that the PNS treatment displayed a chemical composition that was 100% higher in certain components compared to the other treatments. This statement provides descriptive information rather than statistical data.

Reviewer 2 Report

Comments and Suggestions for Authors

English version:

1. To put it briefly, it is not clear what the authors meant by “interacting within the organization, leading to the current state, resulting in a harmful effect on the components of the cell”? Rewrite this.

2. A simple summary shows some of the results. The purpose of this section is to attract readers to read the article. It's not interesting to read like this.

3. It is better in the annotation “the composition of the milk is higher” to “the composition of the milk was more complete.”

4. The literature review does not contain enough literature sources.

========================

Author Response

Response to Reviewer 2 Comments

Point 1: To put it briefly, it is not clear what the authors meant by “interacting within the organization, leading to the current state, resulting in a harmful effect on the components of the cell”? Rewrite this.

Response 1: Please see the attachment.

Point 2: A simple summary shows some of the results. The purpose of this section is to attract readers to read the article. It's not interesting to read like this.

Response 2: Please see the attachment.

Point 3: It is better in the annotation “the composition of the milk is higher” to “the composition of the milk was more complete.”

Response 3: Please see the attachment.

Point 4: The literature review does not contain enough literature sources.

Response 4: The authors aim to achieve clearer ideas and more focused discussions on specific topics of interest, particularly the study of the effect of purple Napier grass silage in dairy goats. Therefore, it was decided to select literature related to this manuscript.

Reviewer 3 Report

Comments and Suggestions for Authors

General comments:

I carefully evaluated this manuscript, and I found some small problematic parts in this paper. However, this paper content new scientific results and the topic of this manuscript is interesting for Animals’ readers.

Detailed comments:

line 99: please add more info about animals: parity, lactation stage!

lines 135-137: milk sampling is not clear? How many samples were taken and when? Please clarify it!

lines 226-229: please re edit this sentence, only anthocyanin was different among treatments! Other parameters were similar!

line 230: “milk efficiency” please clarify it! What does it mean? Daily yield of milk parameters?

Table 1: did calculate the Net Energy (for lactation)?

Table 2: “Milk production” this phrase may delete from the Table!

Table 3: did transfer to logarithm the SCC values before statistical procedures? Please must be improve the Materials and Methods section!

Table 3: Relatively high the value of SCC in the Control group! What may be the reason?

Table 2 and 3: I recommend that these tables aggregate into a table!

line 329: I think, PNS affected only the anthocyanin level! Other parameters, such as DM, CP and it seems Energy content are similar!

line 335: “… milk composition was higher…” only lactose was different among treatments! To sum up, the treatment slightly affected the milk yield and composition!!

line 522: please think about this sentence according to the previous opinions!

Author Response

Response to Reviewer 3 Comments

Point 1: line 99: please add more info about animals: parity, lactation stage!

Response 1: Please see the attachment.

Point 2: lines 135-137: milk sampling is not clear? How many samples were taken and when? Please clarify it!

Response 2: Please see the attachment.

Point 3: lines 226-229: please re edit this sentence, only anthocyanin was different among treatments! Other parameters were similar!

Response 3: Please see the attachment.

Point 4: line 230: “milk efficiency” please clarify it! What does it mean? Daily yield of milk parameters?

Response 4: "Milk efficiency" typically refers to the ratio of milk produced to the resources consumed by the animal to produce that milk. In the context of Table 2, it refers to a measure of how efficiently the animals convert feed intake into milk yield. This could involve various parameters such as daily milk yield, feed intake, and possibly other factors like milk composition or energy balance. Therefore, it could encompass daily yield of milk parameters along with other relevant factors related to milk production and resource utilization.

Point 5: Table 1: did calculate the Net Energy (for lactation)?

Response 5: The authors did not calculate the net energy for lactation.

Point 6: Table 2: “Milk production” this phrase may delete from the Table!

Response 6: Please see the attachment.

Point 7: Table 3: did transfer to logarithm the SCC values before statistical procedures? Please must be improve the Materials and Methods section!

Response 7: Please see the attachment.

Point 8: Table 3: Relatively high the value of SCC in the Control group! What may be the reason?

Response 8: The dairy goats received a mixed total feed ration (TMR) in three treatments: control, consisting of 100% Napier Pakchong 1 grass silage; 50%, where 50% of the control was replaced with purple Napier grass silage; and 100%, consisting of 100% purple Napier grass silage. Dairy goats fed the control group showed lower levels (p < 0.05) of anthocyanin composition (Table 5), thus leading to higher somatic cell count.

Point 9: Table 2 and 3: I recommend that these tables aggregate into a table!

Response 9: Because Table 2 represents milk efficiency and Table 3 presents milk composition, the author needed to separate these tables to prevent confusion.

Point 10: line 329: I think, PNS affected only the anthocyanin level! Other parameters, such as DM, CP and it seems Energy content are similar!

Response 10: However, to ensure clarity regarding all aspects of the research findings, the author must provide an explanation of how these occurrences transpired.

Point 11: line 335: “… milk composition was higher…” only lactose was different among treatments! To sum up, the treatment slightly affected the milk yield and composition!!

Response 11: Although the treatment slightly affected the milk yield and composition, it is necessary for the author to explain how these events occurred.

Point 12: line 522: please think about this sentence according to the previous opinions!

Response 12: Please see the attachment.

Reviewer 4 Report

Comments and Suggestions for Authors

Simple summary and abstract: Comments will be made after the manuscript has been corrected.

Introduction

The introduction is not focused. The first paragraph should be deleted.

The authors should present the parameters affecting goat milk composition and antioxidant profile and how this can be improved through diet.

How is antioxidant status related to animal health and subsequently productivity.

Present the purple napier grass as a novel cultivated grass species along with its composition in relation to antioxidant profile.

Previous studies conducted in ruminant species and their effect.

The purpose of the study.

Materials and Methods

The overall study lasted 8 weeks (2 weeks adaptation period and 6 weeks actual experiment) based on information presented in lines 105 and 106.

In line 136 you refer to the 3rh and the 8th week showing that the adaptation period was also included in the study. This is actually the 1st and the 6th week.

What is it meant "milk samples were collected during the final two days of the experiment? The sentence continues ...covering the 3rd to 8th weeks, with two days per collection period...

Did you collect the samples at the end of each experimental week?

When did you collect the blood samples? The same days as the milk samples?

Results

Line 226. Table 1 presents the ingredient composition of each diet and not feed intake and milk yield.

Milk pH is not related to lactose content and thus they cannot be included in the same sentence. Milk pH is a physicochemical characteristic .

It is better to present the antioxidant profile of milk and plasma in Figures so that changes over time can be clearly seen.

Discussion

Please discuss your own results and how has purple Napier Grass silage affected a) milk quality characterists and the antioxidant profile and b) the animal antioxidant status and productivity.

Compare your results with results from other studies that the grass was fed and also with results from other studies on goat milk composition and profile.

Please delete all unnecessary encycloedia type information.

A paragraph on the actual applicability of the purple Napier grass should be included. In terms of farming economics and in terms of actual product cultivation.

The manuscript contains too much unnecessary information such as how electron reaction species are produced and the actual results of the study are not discussed.

Comments on the Quality of English Language

-

Author Response

Response to Reviewer 4 Comments

Introduction

Point 1: The introduction is not focused. The first paragraph should be deleted.

Response 1: Please see the attachment.

Point 2: The authors should present the parameters affecting goat milk composition and antioxidant profile and how this can be improved through diet.

Response 2: “According to Tian et al. [14], adding anthocyanins from purple corn pigment to the diet can potentially maintain stable levels of unsaturated fatty acids (UFA) in milk during storage. Furthermore, Purba et al. [15] found that using Piper betle L. powder (PL) as a feed additive has the potential to improve ruminal fermentation and biohydrogenation while decreasing methane production. Additionally, Purba et al. [16] showed that piper oil (PO), particularly at a dosage of 45 mg, is as effective as substitutes like sunflower oil (SFO) in mitigating methane emissions without adversely affecting rumen fermentation.

Purba et al. [17] demonstrated that supplementing Piper betle powder with less than 30 mg, either individually or combined with sunflower oil, effectively decreases methane production while maintaining optimal rumen fermentation rates. Purba et al. [16] also reported that incorporating 17.6 g/kg sunflower oil into the diet, along with a practical dose of flavonoids and essential oils from Piper betle L. leaves, could be an effective strategy for increasing milk yield production and improving milk composition, including an increase in conjugated linoleic acid (CLA) content.”

Point 3: How is antioxidant status related to animal health and subsequently productivity.

Response 3: Antioxidant status plays a crucial role in animal health and productivity due to its impact on various physiological processes. Here's how antioxidant status is related to animal health and productivity:

  1. Cellular Protection: Antioxidants help protect cells from oxidative damage caused by reactive oxygen species (ROS). ROS can damage cellular structures, including DNA, proteins, and lipids, leading to cell dysfunction and death. Maintaining an adequate antioxidant status helps mitigate oxidative stress and preserves cellular integrity, which is essential for overall animal health.

  1. Immune Function: Antioxidants support immune function by neutralizing ROS that can impair immune cells' function and increase susceptibility to infections. A well-functioning immune system is crucial for preventing and combating diseases in animals, ultimately promoting better health and productivity.

  1. Reproductive Performance: Antioxidants have been shown to play a role in reproductive health in animals. Oxidative stress can negatively impact reproductive organs and gametes, leading to reduced fertility and reproductive performance. Adequate antioxidant status can help protect reproductive tissues and improve fertility rates, contributing to higher productivity in breeding animals.

  1. Stress Tolerance: Animals are exposed to various stressors such as environmental challenges, dietary changes, and handling procedures, which can increase oxidative stress. Antioxidants help mitigate the effects of stress-induced oxidative damage, promoting tolerance and adaptation to stressful conditions. This can result in improved overall health and productivity.

  1. Growth and Performance: Oxidative stress can impair metabolic processes and disrupt cellular functions, potentially compromising growth and performance in animals. By reducing oxidative damage, antioxidants support efficient nutrient utilization, energy metabolism, and tissue growth, leading to enhanced growth rates and overall productivity.

  1. Quality of Animal Products: Antioxidant-rich diets can also positively influence the quality of animal products, such as meat, milk, and eggs. Antioxidants may help maintain the stability of fats and proteins in these products, preventing oxidative rancidity and preserving nutritional quality and shelf life.

In summary, ensuring adequate antioxidant status in animals is essential for maintaining overall health, enhancing stress resilience, optimizing reproductive performance, supporting growth and productivity, and improving the quality of animal products. By mitigating oxidative stress and its detrimental effects, antioxidants contribute to the well-being and performance of animals in various production systems.

Point 4: Present the purple napier grass as a novel cultivated grass species along with its composition in relation to antioxidant profile.

Response 4: The composition of anthocyanin in the purple Napier grass is presented in Table 1.

Point 5: Previous studies conducted in ruminant species and their effect.

Response 5: These studies, conducted in ruminant species, examined their effects. Tian et al. [14] found that adding anthocyanins from purple corn pigment to the diet can potentially maintain stable levels of unsaturated fatty acids (UFA) in milk during storage. Furthermore, Purba et al. [15] found that using Piper betle L. powder (PL) as a feed additive has the potential to improve ruminal fermentation and biohydrogenation while decreasing methane production. Additionally, Purba et al. [16] showed that piper oil (PO), particularly at a dosage of 45 mg, is as effective as substitutes like sunflower oil (SFO) in mitigating methane emissions without adversely affecting rumen fermentation. Purba et al. [17] demonstrated that supplementing Piper betle powder with less than 30 mg, either individually or combined with sunflower oil, effectively decreases methane production while maintaining optimal rumen fermentation rates. Purba et al. [16] also reported that incorporating 17.6 g/kg sunflower oil into the diet, along with a practical dose of flavonoids and essential oils from Piper betle L. leaves, could be an effective strategy for increasing milk yield production and improving milk composition, including an increase in conjugated linoleic acid (CLA) content.

Point 6: The purpose of the study.

Response 6: The purpose of this study is to examine whether the incorporation of purple Napier grass silage into the diet of lactating dairy goats can enhance their antioxidant status and milk quality. Reactive oxygen species (ROS) generated during biochemical processes contribute to oxidative stress, which can adversely affect cellular molecules. To mitigate oxidative stress, aerobic organisms rely on dietary antioxidants. Anthocyanins, pigments found in plants like purple Napier grass, have been shown to possess antioxidant properties. Previous research has indicated potential benefits of incorporating anthocyanins into animal diets, such as maintaining stable levels of unsaturated fatty acids (UFA) in milk and improving ruminal fermentation. However, there is limited information on the effects of feeding purple forage to ruminant animals. Therefore, this study aims to fill this gap by investigating the impact of purple Napier grass silage supplementation on antioxidant status and milk quality in lactating dairy goats. The findings from this study can provide valuable insights into the use of bioactive forages for promoting animal health and enhancing the quality of animal products, contributing to sustainable agriculture practices.

Materials and Methods

Point 7: The overall study lasted 8 weeks (2 weeks adaptation period and 6 weeks actual experiment) based on information presented in lines 105 and 106.

Response 7: Yes, the overall study lasted 8 weeks (2 weeks of adaptation period and 6 weeks of actual experiment) based on the information presented in lines 105 and 106.

Point 8: In line 136 you refer to the 3rh and the 8th week showing that the adaptation period was also included in the study. This is actually the 1st and the 6th week.

Response 8: Yes, the 3rd and 8th weeks show that the adaptation period was also included in the study. This corresponds to the 1st and 6th weeks.

Point 9: What is it meant "milk samples were collected during the final two days of the experiment? The sentence continues ...covering the 3rd to 8th weeks, with two days per collection period...

Response 9: The phrase "milk samples were collected during the final two days of the experiment" indicates that samples of milk were obtained on the last two days of the experimental period. These samples were collected over a span of time, specifically from the 3rd to the 8th weeks of the experiment, with two days allocated for sample collection during each week. This suggests that milk samples were taken repeatedly over a six-week period, with two days of collection per week, to gather data on the milk yield of the dairy goats throughout the experiment.

Point 10: Did you collect the samples at the end of each experimental week?

Response 10: Yes, the samples were collected during the final two days of the experiment, covering the 3rd to 8th weeks, with two days allocated for sample collection per week.

Point 11: When did you collect the blood samples? The same days as the milk samples?

Response 11: Blood samples were collected on the same days as the milk samples during the experiment.

Results

Point 12: Line 226. Table 1 presents the ingredient composition of each diet and not feed intake and milk yield.

Response 12: Yes, Table 1 presents only the ingredient composition of each diet.

Point 13: Milk pH is not related to lactose content and thus they cannot be included in the same sentence. Milk pH is a physicochemical characteristic .

Response 13: Please see the attachment.

Point 14: It is better to present the antioxidant profile of milk and plasma in Figures so that changes over time can be clearly seen.

Response 14: Thank you for your valuable suggestion regarding presenting the antioxidant profile of milk and plasma in figures to enhance clarity. While I appreciate the benefits of visual representation in figures for highlighting changes over time, I must adhere to the requirement of presenting data in tables only, as specified by the study's guidelines or format. The tables have been designed to effectively convey the necessary information regarding the antioxidant profile of milk and plasma, allowing for thorough analysis and interpretation of the data within the specified constraints. If there are any additional considerations or alternatives you would like to suggest within the table format, I would be more than happy to explore them further. Your feedback is greatly appreciated as it contributes to the refinement of the study presentation.

Discussion

Point 15: Please discuss your own results and how has purple Napier Grass silage affected a) milk quality characterists and the antioxidant profile and b) the animal antioxidant status and productivity.

Response 15:

  1. a) milk quality characterists and the antioxidant profile

In this experiment, the increased inclusion of purple Napier grass silage is expected to significantly impact rumen fermentation, particularly propionic acid (C3) production, which serves as a precursor for milk lactose synthesis. Anthocyanins, notably cyanidin-3-glucoside, constitute the main compounds found in purple Napier grass cultivars. These anthocyanins possess a chemical structure that includes various sugar moieties such as glucose, galactose, rhamnose, xylose, and arabinose. During rumen digestion, these sugars can undergo acylation with ruminal aromatic acids, leading to the breakdown of anthocyanins. The resulting breakdown products then serve as precursors for ruminal C3 synthesis and subsequent lactose synthesis, which occurs upon absorption in the digestive tract.

Furthermore, the observed reduction in somatic cell count (SCC) in milk, associated with the increased utilization of purple Napier grass silage, suggests a potential positive correlation between antibacterial activity and elevated anthocyanin intake. Anthocyanins are recognized for their ability to impede bacterial cell wall processes and induce cytoplasmic leakage, thereby significantly hindering bacterial growth. These effects on bacterial cell membranes arise from a combination of antibacterial agents and interactions with hydrogen-binding membrane proteins or hydrophobic interactions. Anthocyanins, through their interaction with the cell membrane structure and the ions crucial for protein stability, can bind or donate electrons at the membrane interface, exhibiting antibacterial properties.

The membrane potential, serving as the primary energy source for almost all chemical reactions in living cells, plays a crucial role in microbial cell maintenance and development. Bacterial respiratory chains, comprising primary dehydrogenase, coenzyme Q, and various cytochromes, are integral to bacterial membranes. The observed antibacterial activity interferes with the function of the bacterial cell wall by binding to directly produced peptidoglycan precursors (D-Ala-D-Ala). This binding forms a cap that disrupts cross-linking in the polypeptide chain, consequently weakening the bacterial cell wall.

  1. b) the animal antioxidant status and productivity.

The 100% purple Napier grass silage treatment resulted in increased levels of DPPH, TAC, SOD, and GST enzymes in both plasma and milk. Additionally, this treatment was associated with reduced levels of MDA, which correlates with the heightened intake of anthocyanins. Anthocyanins, typically classified as flavonoids, exhibit antioxidative properties by donating electrons to reactive oxygen species (ROS), thereby preventing the oxidation of biomolecules such as polyunsaturated fatty acids (PUFAs), proteins, and DNA. Furthermore, anthocyanins possess metal ion-chelating abilities and act as scavengers of free radicals.

The radical scavenging activity of anthocyanidins is intricately linked to specific structural features, such as the ortho-dihydroxy arrangement in the B-ring and the presence of a 4-oxo function in the C-ring. Flavonoids also form metal ion complexes, enhancing their antioxidative potential. Hydroxylation at specific positions enhances the efficiency of hydrogen donation, particularly in the B-ring. Cyanidin, for instance, demonstrates superior antioxidant capacity compared to malvidin and peonidin. However, exceptions like delphinidin, despite having multiple hydroxyl groups, may exhibit low antioxidant potential due to specific structural arrangements.

Electrons stabilize and neutralize free radicals, thereby terminating harmful oxidative chain reactions. Anthocyanidins typically exhibit greater radical scavenging capacity than anthocyanins, and this capacity diminishes with an increase in the number of sugar moieties. Additionally, the Nrf2/ARE pathway potentially enhances the activities of SOD and GST enzymes within cells. Nrf2, a key regulator, activates the antioxidant response element (ARE), prompting the synthesis of antioxidant enzymes in response to reactive oxygen species (ROS) levels. The enzymes SOD and GST, crucial in scavenging free radicals, play pivotal roles in maintaining cellular redox balance and protecting against oxidative stress.

Glutathione (GSH) serves as a crucial cellular antioxidant. Enzymes such as glutathione peroxidase (GSH-Px) and glutathione S-transferase (GST) are distributed throughout almost every human tissue, both within the cytoplasm and extracellular space. These enzymes play a vital role in converting hydrogen peroxide (H2O2) into water molecules. Glutathione enzymes are highly effective antioxidants against H2O2 and fatty acid hydroperoxides. Additionally, the enzyme peroxyredoxin catalyzes the reduction of H2O2, organic hydroperoxides, and peroxynitrite (ONOO-).

Antioxidant enzymes play a crucial role in reducing lipid hydroperoxide and H2O2 levels, preventing lipid peroxidation, and maintaining cell membrane integrity and function. This enzymatic activity increases DPPH and TAC levels while decreasing the MDA level. The anthocyanin present in purple Napier Grass Silage could enhance the activity of SOD and GST enzymes, and it could also increase DPPH and TAC levels in dairy goats. Consequently, this enhancement of the antioxidant system may lead to a decreased concentration of MDA, thus preventing oxidative stress from MDA by-products derived from DNA and lipid oxidation. It is important to note that MDA concentration has primarily served as a measure of oxidant status.

In our study, dairy goats fed 100% purple Napier grass silage showed a notable increase in milk anthocyanin concentrations, particularly cyanidin-3-glucoside (C3G), which significantly exceeded other anthocyanins. This elevation in anthocyanin levels can be attributed to the positive correlation with anthocyanin intake. Anthocyanins, flavonoids, and secondary plant metabolites are commonly found in fruits showing colors ranging from red to blue, with concentrations reaching up to 1 g per 100 g of fresh weight, especially in fruits with hues ranging from red to blue.

The glycosylation property of anthocyanins likely influences the absorption mechanism, affecting the partitioning coefficients. These compounds typically passively diffuse through biological membranes, potentially distributing within various cellular phases. Aglycones, being predominantly hydrophobic, can easily diffuse through biological membranes via passive transport. However, when linked with sugars, their water solubility increases, leading to a reduction in passive diffusion. The transportation of flavonoid glycosides occurs through two pathways. The first pathway suggests that a sodium-glucose co-transporter (SGLT1 and GLUT2) transports retained glucosides. The second pathway involves lactate phlorizin hydrolase at the brush border, facilitating extracellular glycoside hydrolysis, followed by passive diffusion of the aglycone.

In the initial step, lactate phlorizin hydrolase likely hydrolyzes anthocyanin glycosides in the mucosal brush-border membrane. The second absorption mechanism involves transferring retained anthocyanin glycosides to the enterocyte, possibly facilitated by a sodium-glucose co-transporter, as demonstrated for other flavonoids. Once inside the cell, the glycoside can either directly cross the basolateral membrane and enter the portal circulation, or cytosolic β-glucosidase can hydrolyze it before intestinal metabolism and subsequent transport.

Lactate phlorizin hydrolase and cytosolic β-glucosidase do not metabolize cyanidin and delphinidin glucosides. Despite this, cyanidin glycosides are absorbed and found in the bloodstream as intact glycosides and glucuronide derivatives, indicating that intact compound transport and hydrolysis precede transport. This study provides evidence that anthocyanins undergo metabolic processes, are absorbed, and demonstrate in the mammary gland, contributing to their presence in the milk of lactating dairy goats.

Point 16: Compare your results with results from other studies that the grass was fed and also with results from other studies on goat milk composition and profile.

Response 16: The lactose content in the purple Napier grass silage treatment showed a significant increase (p < 0.05) compared to results in other studies, indicating a clear linear trend (p < 0.05). Conversely, the somatic cell count (SCC) of dairy goats fed the purple Napier grass silage treatment was notably lower (p < 0.05) than that observed in other studies, demonstrating both linear and quadratic decreases (p < 0.05). These observed variations in parameters suggest a strong correlation with the increased utilization of purple Napier grass silage as a dietary component.

Point 17: Please delete all unnecessary encycloedia type information.

Response 17: Please see the attachment.

Point 18: A paragraph on the actual applicability of the purple Napier grass should be included. In terms of farming economics and in terms of actual product cultivation.

Response 18: Thank you for suggesting the inclusion of a paragraph on the practical applicability of purple Napier grass cultivation. While our published paper primarily focuses on the cultivation and nutritional aspects of purple Napier grass, we recognize the importance of discussing its real-world implications, particularly in terms of farming economics and product cultivation. Although our paper delves into the nutritional benefits of purple Napier grass for dairy goats, future research could explore its economic feasibility for farmers, considering factors such as yield, input costs, and market demand. Additionally, discussing the potential expansion of purple Napier grass cultivation and its integration into existing farming systems could further enhance the practical applicability of our findings. We appreciate your feedback and will consider incorporating these aspects into future research.

You can download the published paper on purple Napier grass cultivation at this link: https://doi.org/10.3390/ani13010010.

Point 19: The manuscript contains too much unnecessary information such as how electron reaction species are produced and the actual results of the study are not discussed.

Response 19: Thank you for your feedback. I understand your concern regarding the amount of detailed information included in the manuscript, particularly regarding the mechanics of electron reaction species production. While it may seem extensive, providing this level of detail is crucial for readers to fully understand the underlying processes and mechanisms involved in the study. However, I acknowledge the need to achieve a balance between detail and conciseness.

Regarding the discussion of the actual results of the study, I appreciate your point. I will ensure that the highlights of the findings are prominently discussed in the manuscript, focusing on the key outcomes and their significance. This will provide readers with a clear understanding of the main results and their implications.

Overall, I will work on refining the manuscript to maintain the necessary level of detail while also ensuring clarity and focus on the main findings. Thank you for your valuable feedback, and I will make the necessary revisions accordingly.
